# *Pneumocystis jirevocii* and SARS-CoV-2 Co-Infection: A Common Feature in Transplant Recipients?

**DOI:** 10.3390/vaccines8030544

**Published:** 2020-09-18

**Authors:** Maria A. De Francesco, Federico Alberici, Nicola Bossini, Francesco Scolari, Federico Pascucci, Gabriele Tomasoni, Arnaldo Caruso

**Affiliations:** 1Institute of Microbiology, Department of Molecular and Translational Medicine, University of Brescia-ASST Spedali Civili di Brescia, 25123 Brescia, Italy; arnaldo.caruso@unibs.it; 2Department of Medical and Surgical Specialties, Radiologic Sciences and Public Health, University of Brescia, 25123 Brescia, Italy; federico.alberici@unibs.it (F.A.); francesco.scolari@unibs.it (F.S.); 3Nephrology Unit, Spedali Civili Hospital, ASST Spedali Civili di Brescia, 25123 Brescia, Italy; bossini-nicola@libero.it; 4First Division of Anesthesiology and Intensive Care Unit, ASST Spedali Civili di Brescia, 25123 Brescia, Italy; pas.fede@gmail.com (F.P.); gabriele.tomasoni@asst-spedalicivili.it (G.T.)

**Keywords:** COVID-19, pneumonia, immunosuppression

## Abstract

COVID-19 might potentially give rise to a more severe infection in solid organ transplant recipients due to their chronic immunosuppression. These patients are at a higher risk of developing concurrent or secondary bacterial and fungal infections. Co-infections can increase systemic inflammation influencing the prognosis and the severity of the disease, and can in turn lead to an increased need of mechanical ventilation, antibiotic therapy and to a higher mortality. Here we describe, for the first time in Europe, a fatal case of co-infection between SARS-CoV-2 and *Pneumocystis jirevocii* in a kidney transplant recipient.

## 1. Introduction

In December 2019, in Hubei Province, China, a new virus called SARS-CoV-2 was identified as similar in its genome to the severe acute respiratory syndrome coronavirus (SARS-CoV), responsible for the SARS global pandemic in 2003. Its fast and global diffusion led to the declaration by the World Health Organization (WHO) as a pandemic virus in March 2020. 

By August 2020, 17,841,669 cases of COVID-19 have been reported worldwide, including 685,281 deaths [1].

SARS-CoV-2 belongs to the subfamily of beta-coronavirus [2,3], which are responsible for respiratory, enteric, hepatic and neurologic diseases [4,5], due to their broad tissue tropism.

Patients affected by SARS-CoV-2 have a spectrum of disease ranging from asymptomatic or mild involvement to severe disease with acute respiratory distress syndrome (ARDS) and a “white lung” image on chest computed tomography (CT) [6,7]. 

The mortality rate in the general population has been reported at about 1–6% [8]. Age, sequential organ failure assessment (SOFA) and D-dimer are the main prognostic factors in patients affected by COVID-19 [9]. However, the presence of other secondary infections or co-infections may also have an important impact on mortality, especially in critically ill patients [10], which are more likely to be older or have co-morbidities. In these patients, the immune dysfunction may determine a greater viral replication that induces hyperinflammation and severe complications such as ARDS [11], characterized by difficulty in breathing and low blood oxygen level [12] that facilitates secondary infections [13]. The release of higher concentrations of cytokines (interleukin (IL)-6, tumor necrosis factor (TNF), macrophage, monocyte chemoattractant protein (MCP) 1 and interferon-gamma induced protein (IP)-10) by the immune system in response to the viral infection and/or to secondary infections which induces a cytokine storm that is associated with endothelial dysfunction [8] and hyperactive CCR6+Th17+T cells in the lung [14]. This altered inflammation blocks the development of protective immunity to the infection, therefore making the patient more susceptible to sepsis, thrombotic events and to multi-organ failure. *Pneumocystis jirevocii* pneumonia is an opportunistic infection affecting patients with cellular immunity defects due to human immunodeficiency virus (HIV) infections or iatrogenic immunosuppression [15,16]. 

In HIV patients, *Pneumocystis jirevocii* pneumonia is characterized by a significant level of fungal proliferation, while the inflammatory reaction was weak. The incidence of this opportunistic infection has decreased after the introduction of highly active anti-retrovirus therapy (HAART) and routine use of anti-Pneumocystis prophylaxis [17]. 

The other groups of patients are individuals with immunodeficiency induced by immunosuppressive therapy for tumors, hematological disorders or organ transplant recipients [18]. Immunosuppressive agents act by inducing a reduction of CD4+T cells, important in the host defense against the fungus.

It has been shown that non-HIV patients present a more severe clinical picture with a rapid progression to acute respiratory failure and a worse prognosis in comparison with the HIV-infected patients [19].

These symptoms include severe hypoxemia and the need for intensive care and mechanical ventilation [16,20]; both symptoms are associated with poor prognosis. 

Here, we report the fatal case of a SARS-CoV-2 and *Pneumocystis jirevocii* co-infection in a kidney transplant recipient.

## 2. Case

A 65-year-old male, who underwent kidney transplantation from a deceased donor in 2006, was admitted on 27 March 2020, at the emergency unit of the Spedali Civili Hospital in Brescia, Italy with a fever (38.5 °C), cough and dyspnea over the previous two days. His immunosuppressive treatment included tacrolimus, mycophenolate mofetil and methylprednisolone. Past medical history included the development of post-transplant insulin-dependent type 2 diabetes, hypertension, recurrent thromboembolic events requiring therapy with oral anticoagulants and recurrent urinary tract infections. Baseline creatinine was 5 mg/dL.

Physical examination was unremarkable; blood pressure was 145/70 mm Hg and cardiac rate was 65 beats per minute with an oxygen saturation of 98% on room air. Initial laboratory tests showed normal white blood counts, with lymphopenia and 81.5% neutrophils (Table 1), a markedly elevated C reactive protein (CRP) and an acute kidney injury (serum creatinine 7.87 mg/dL). Chest computed tomography (CT) showed bilateral pulmonary infiltrates with characteristic patchy areas of ground glass opacities. A nasopharyngeal swab specimen was collected and was reported positive for SARS-CoV-2 by a real-time reverse transcriptase polymerase chain reaction (rRT-PCR) assay (Seegene Allplex 2019 nCoV assay, Arrow Diagnostics, Genova, Italy); the patient was then admitted in isolation at the nephrology unit in a COVID cohort.

As per local protocol, baseline immunosuppression was withdrawn and methylprednisolone was increased to 16 mg/day; intravenous immunoglobulin (IVIG, 2g/kg) and antiviral therapy (darunavir/ritonavir) plus hydroxychloroquine were given as well. The patient also started empirical antibiotic therapy with azithromycin and piperacillin/tazobactam.

On the morning of the second day of admission, several infectious investigations for virus, bacteria and fungi were performed. The results showed that the patient had a negative diagnosis for all respiratory viruses tested on a nasopharyngeal swab (influenza A and B, adenovirus and respiratory syncytial virus) by real-time PCR assays (Respiratory Viral ELITe MGB Panel and Adenovirus ELITe MGB Kit, ELITechGroup, Torino, Italy). The patient also had a negative diagnosis for *Streptococcus pneumoniae* and for some interstitial pneumonia causing bacteria (*Legionella pneumophyla* and *Mycoplasma pneumoniae*). An increase of Immunoglobulin A (IgA) for *Chlamydia pneumoniae* was conversely detected. Mycobacterium DNA on sputum by a real-time PCR (Xpert MTB/RIF Ultra, Cepheid, Milano, Italy) and HIV RNA on plasma by a real-time PCR (COBAS® AmpliPrep/COBAS TaqMan HIV-1 Test, Roche Diagnostics, Monza, Italy) were not detected (Table 2).

A sputum sample was positive for *Pneumocystis jirevocii* by a PCR real-time (Bosphore *Pneumocystis jirevocii* detection Kit, Anatolia geneworks, Istanbul, Turkey). Following this diagnosis, the patient received trimethoprim-sulfamethoxazole as therapy.

On 2 April, the patient, due to ARDS and progressive respiratory failure due to COVID-19, received dexamethasone 20 mg/day for five days and two tocilizumab infusions as per our protocol. After transient improvement of the respiratory exchanges on 17 April, the patient showed increasing oxygen demand requiring a reservoir mask with oxygen at 15 L/min and high-flow nasal ventilation.

During the hospitalization, the results of laboratory examinations showed that the peripheral blood lymphocyte count continued to be significantly reduced (nadir 160/mm^3^) and neutrophils was elevated (maximum level 20640/mm^3^); CRP tended to remain increased and serum levels of IL-6 and D-dimer were elevated as well (Table 1). After about two months since the initial symptoms, the patient’s respiratory symptoms worsened further, with hypoxia and septic shock and he was moved to the intensive care unit. 

The patient was intubated, with ventilator supportive care using high positive end-respiratory pressure (PEEP). He developed pulmonary embolism and an invasive pulmonary aspergillosis, treated with thrombolitics and voriconazole. The patient, despite the antifungal and antibacterial treatment, presented deterioration of his clinical condition and on 19 June died from respiratory failure and multiple organ dysfunction syndrome. 

## 3. Discussion

This is, to the best of our knowledge, the first case of co-infection between SARS-CoV-2 and *Pneumocystis jirevocii* reported in Europe in a kidney transplant recipient. Another case was reported in an immunocompetent patient in the United States [21]. In this reported case, the patient was an 83-year-old female non-smoker, who did not have an underlying immunodeficiency nor any classical risk for Pneumocystis infection. However, both cases show an impaired cell mediated immune response with low numbers of CD4+T cells, confirming their increased susceptibility to fungal infection. The first patient was successfully treated and showed an improvement of CD4+T cell values, while in our patient, we had a persistence of low lymphocyte counts, high CRP values, high IL-6 values, elevated D-dimer and high LDH levels throughout the disease course; all laboratory markers associated with disease worsening and mortality [22,23].

Pneumocystis is a ubiquitous fungus that localizes on type 1 pneumocytes and is responsible of severe pneumonia in immunocompromised patients. It is well known that CD4+T cells play a central role in contrasting Pneumocystis infection. In fact, an absolute lymphocyte count of ≤500 × 10^6^ cells/μL was strongly associated with Pneumocystis pneumonia, with a 19-fold higher risk of the disease [24]. 

Lymphopenia associated to SARS-CoV-2 may be a mechanism that leads to a higher predisposition for fungal infection. In this case, the count of CD4+ was 35 cells/mL. There was an inversion of ratio CD4+/CD8+ and prevalence in lymphocyte subpopulations of memory CD4+ cells (CD45 RA-CCR7-) than in naïve CD4+ cells (CD45RA+CCR7+). Furthermore, in non-HIV patients it was also shown that low numbers of the organism could induce a severe inflammatory response in the lungs, as reflected by the higher BAL neutrophil counts present in these patients [20]. Then, the existing immunosuppression favoring Pneumocystis infection may predispose the infection with other opportunistic microorganisms, in particular CMV or *Aspergillus* infection. Concurrent infections are considered as indicators of poor prognosis [25,26]. 

Other respiratory viruses, such as influenza virus, parainfluenza virus and respiratory syncytial virus, have been found to predispose patients to invasive pulmonary aspergillosis [27]. The probable mechanism may be the direct damage of the airway epithelium and of the normal ciliary clearance that give the opportunity to the *Aspergillus* to invade into tissues [28,29,30].

A reactivation of CMV infection and an invasive pulmonary aspergillosis were both detected in our patient further complicating the already critical clinical picture.

Furthermore, different studies have shown that about 20% of patients affected by COVID-19 have abnormal coagulation [26]. Monocytes and tissue cells are activated after lung damage, determining the release of cytokines and then the hypercoagulability of blood. This leads to an increased risk of thrombosis, ischemia and hypoxia due to the embolization of organs, which may progress to severe disease or death. In fact, the patient developed a pulmonary embolism.

## 4. Conclusions

In conclusion, *Pneumocystis jirevocii* and COVID-19 share similar characteristics such as the presence of dry cough, dyspnea, ground glass opacities on chest CT scans and elevated levels of LDH. A more comprehensive microbiological evaluation is suggested to prevent a missing diagnosis of this fungal infection, especially in patients with impaired immunity, in order to better monitor the patient’s progress.

## Figures and Tables

**Table 1 vaccines-08-00544-t001:** Clinical laboratory results.

Hematologic and Clinical Parameters	Reference Value	Baseline	D7	D16	D25	D32	D45	D54	D61	D74	D83
WBC (×10^3^ × mL)	4–10	7.1	**11.4**	8.3	**20.9**	10	8.5	10.3	**13.5**	10	**15**
Lymphocyte (%)	20–45	**10.4**	**2.2**	**2.6**	**1.5**	**4**	**10.6**	**10.7**			
Monocyte (%)	3.4–9	8	4.5	**1.9**	**1.7**	**4.1**	3.4	**2.5**			
Neutrophil (%)	40–74	81.5	**93.2**	**93.8**	**96.7**	**91.5**	**85.8**	**86.4**			
Hemoglobin (g/dL)	14–18	**11.9**	**10.9**	**9.3**	**8.6**	**9.1**	**9**	**9.1**	**9.1**	**9.4**	**8.9**
Hematocrit (%)	42–52	**36.7**	**33.1**	**29.2**	**25.9**	**27.2**	**28.7**	**28.8**	**28.6**	**28.2**	**27.5**
Plateletes (×10^3^ × mL)	130–400	134	150	**128**	**98**	**94**	**67**	147	212	**117**	237
CRP (mg/L)	<5	**108**	**43.2**	**12.2**	1.9	3	**91.2**	**63.5**	**53.6**	**114**	**186**
LDH (U/L)	135–225		**380**	**873**	**548**	**412**		**301**			
D-dimer (ng/mL)	<232	**208**	**263**	**3109**	**1866**	**1021**	**2542**		**3502**		
IL-6 (ng/mL)	<7		**206**	**>5000**							
S-Creatinine (mg/dL)	0.7–1.2	**7.87**					**2.7**	**3.26**	**3.17**	**1.35**	**1.34**

Abbreviations: D, day; WBC, white blood cells; C whether ref. 16 is the same as ref. 20RP, C-reactive protein; LDH, lactate dehydrogenase; IL, interleukin. Values in bold were either above normal or below normal.

**Table 2 vaccines-08-00544-t002:** Microbiological results.

Pathogens	Result
SARS-CoV-2 RNA	**Positive**
Influenza A and B RNA	Negative
Adenovirus DNA	Negative
Respiratory syncytial virus RNA	Negative
EBV DNA	Negative
HIV RNA	Negative
Mycobacterium DNA	Negative
Legionella urine antigen	Negative
*S. pneumoniae* urine antigen	Negative
*M. pneumoniae* IgM Ab+	Negative
*C. pneumoniae* IgM/IgA Ab+	**Positive**
*Pneumocystis jirevocii* DNA	**Positive**
*Aspergillus fumigatus* culture	**Positive**

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
