# Peer review of "Pneumocystis jirevocii and SARS-CoV-2 Co-Infection: A Common Feature in Transplant Recipients?"

_vaccines, 2020, doi:10.3390/vaccines8030544_

Round 1

Reviewer 1 Report

 The author should re-organize the part of the introduction.

In this part, I  suggest the author add more information after this sentence.

In fact, in older patients or patients with co-morbidities, the immune dysfunction may determine a greater viral replication which induces hyper inflammation and severe complications such as ARDS, which facilitates secondary infections.

The author could learn more in this paper titled Trained immunity: a tool for reducing susceptibility and severity of SARS-CoV2 infection.

Author Response

The author should re-organize the part of the introduction.

In this part, I  suggest the author add more information after this sentence.

In fact, in older patients or patients with co-morbidities, the immune dysfunction may determine a greater viral replication which induces hyper inflammation and severe complications such as ARDS, which facilitates secondary infections.

The author could learn more in this paper titled Trained immunity: a tool for reducing susceptibility and severity of SARS-CoV2 infection.

Answer:

According to the reviewer’s suggestion, we have added more information.

Reviewer 2 Report

Summary

In this manuscript, a SARS-CoV-2 and Pneumocystis jirevocii co-infection case report is described.

Broad Comments

  1. The Introduction is disjointed. This section needs to be rewritten. Instead of 14 paragraphs, considered writing a few paragraphs that flow together.
  2. By what methods were the microbiological results obtained? More details are needed. Other than the antigen tests listed, were the remainder of the tests PCR-based from the nasopharyngeal swab specimen? Was this in-house PCR or commercial PCR tests?
  3. The authors mention that another case of SARS-CoV-2 and Pneumocystis jirevocii co-infection has already been reported. What is novel about the current study? What was similar between the two cases?

Specific Comments

  1. Title: In the title, should Pneumocystis be italicized?
  2. Introduction (paragraph 1): The virus is SARS-CoV-2 and the disease is COVID-19. There is a distinction between the two that should be made.
  3. Introduction (paragraph 2): What is meant by extensive tropism? This virus seems to have broad tissue tropism.
  4. Case (paragraph 1): Add the year after “March 27”.
  5. Conclusions: In the first sentence, “Pneumocystis jirevocii and COVID-19 share different characteristics such as…”. Instead, should this read, “Pneumocystis jirevocii and COVID-19 share similar characteristics such as…”?

Author Response

Thanks for your valuable comments.

Summary

In this manuscript, a SARS-CoV-2 and Pneumocystis jirevocii co-infection case report is described.

Broad Comments

  1. The Introduction is disjointed. This section needs to be rewritten. Instead of 14 paragraphs, considered writing a few paragraphs that flow together.

Answer:

According to the referee’s suggestion, we tried to shorten the introduction except for the part that has been  extended according to the request of the other referee

  1. By what methods were the microbiological results obtained? More details are needed. Other than the antigen tests listed, were the remainder of the tests PCR-based from the nasopharyngeal swab specimen? Was this in-house PCR or commercial PCR tests?

Answer:

All the tests used were commercial assays. We have specified materials and kits used for PCR assays

  1. The authors mention that another case of SARS-CoV-2 and Pneumocystis jirevocii co-infection has already been reported. What is novel about the current study? What was similar between the two cases?

Answer:

The two cases have been compared.

Specific Comments

  1. Title: In the title, should Pneumocystis be italicized?

Answer:

Pneumocystis has been italicized

  1. Introduction (paragraph 1): The virus is SARS-CoV-2 and the disease is COVID-19. There is a distinction between the two that should be made.

Answer:

The mistake has been corrected

  1. Introduction (paragraph 2): What is meant by extensive tropism? This virus seems to have broad tissue tropism

Answer:

The sentence has been changed

  1. Case (paragraph 1): Add the year after “March 27”.

Answer:

The year has been added

  1. Conclusions: In the first sentence, “Pneumocystis jirevocii and COVID-19 share different characteristics such as…”. Instead, should this read, “Pneumocystis jirevocii and COVID-19 share similar characteristics such as…”?

Answer:

The mistake has been corrected

Reviewer 3 Report

In this paper, authors presented a case report on Pneumocystis jirevocii and SARS-CoV-2 co-infection. This case report is overall well written and described. Few minor revisions from my side:

  1. The abstract is too short. Please elaborate little bit to make it compelling.
  2. "More than 4.5 million of cases were confirmed at the end of May 2020, with more than 300,000 deaths in the world"...... Please update with current case numbers.
  3. A similar case report is published recently: Menon AA, Berg DD, Brea EJ, et al. A Case of COVID-19 and Pneumocystis jirovecii Coinfection. Am J Respir Crit Care Med. 2020;202(1):136-138. doi:10.1164/rccm.202003-0766LE...............Please introduce this one and discuss the comparative findings.
  4. Authors should discuss elaborately some of the data presented at Table 1a, particularly the bold ones.

The case report is acceptable after these minor revision.

Author Response

In this paper, authors presented a case report on Pneumocystis jirevocii and SARS-CoV-2 co-infection. This case report is overall well written and described. Few minor revisions from my side:

  1. The abstract is too short. Please elaborate little bit to make it compelling.

Answer:

The abstract has been revised

  1. "More than 4.5 million of cases were confirmed at the end of May 2020, with more than 300,000 deaths in the world"...... Please update with current case numbers.

Answer:

Case numbers have been updated

  1. A similar case report is published recently: Menon AA, Berg DD, Brea EJ, et al. A Case of COVID-19 and Pneumocystis jirovecii Coinfection. Am J Respir Crit Care Med. 2020;202(1):136-138. doi:10.1164/rccm.202003-0766LE...............Please introduce this one and discuss the comparative findings.

Answer:

The two cases have been compared

  1. Authors should discuss elaborately some of the data presented at Table 1a, particularly the bold ones.

Answer:

We have inserted a sentence in the Discussion section.

The case report is acceptable after these minor revision.

Round 2

Reviewer 2 Report

The authors have addressed my previous comments.